# YM155 Inhibition of Survivin Enhances Carboplatin Efficacy in Metastatic Castration-Resistant Prostate Cancer

**DOI:** 10.3390/ph18111752

**Published:** 2025-11-18

**Authors:** Vicenç Ruiz de Porras, Martin K. Bakht, Maria Fernandez-Saorín, Clara Alcon, Luis Palomero, Júlia Francisco-Rodon, Mariona Figols, Joan Montero, Vincenza Conteduca, Himisha Beltran, Albert Font

**Affiliations:** 1GRET and Toxicology Unit, Department of Pharmacology, Toxicology and Therapeutic Chemistry, Faculty of Pharmacy and Food Sciences, University of Barcelona, 08028 Barcelona, Spain; 2Department of Medical Oncology, Dana-Farber Cancer Institute, Boston, MA 02215, USA; martin_bakht@dfci.harvard.edu (M.K.B.); himisha_beltran@dfci.harvard.edu (H.B.); 3CARE Program, Germans Trias i Pujol Research Institute (IGTP), Camí de les Escoles, s/n, 08916 Badalona, Spain; mfernandezs@igtp.cat (M.F.-S.); jfrancisco@igtp.cat (J.F.-R.); afont@iconcologia.net (A.F.); 4Badalona Applied Research Group in Oncology (B-ARGO), Catalan Institute of Oncology, Camí de les Escoles, s/n, 08916 Badalona, Spain; 5Department of Biomedical Sciences, Faculty of Medicine and Health Sciences, University of Barcelona, 08007 Barcelona, Spain; clara.alcon@ub.edu (C.A.); jmontero@ub.edu (J.M.); 6Department Llenguatges i Sistemes Informàtics, Institute of New Imaging Technologies, Universitat Jaume I, Av. Sos Baynat s/n, 12071 Castellón de la Plana, Spain; lpalomerol@gmail.com; 7Medical Oncology Department, Althaia Xarxa Assistencial Universitària de Manresa, C/Dr. Joan Soler, 1-3, 08243 Manresa, Spain; mfigols@althaia.cat; 8Networking Biomedical Research Center in Bioengineering, Biomaterials and Nanomedicine (CIBER-BBN), 28029 Madrid, Spain; 9Unit of Medical Oncology and Biomolecular Therapy, Department of Medical and Surgical Sciences, University of Foggia, 71122 Foggia, Italy; vincenza.conteduca@unifg.it; 10Department of Medical Oncology, Catalan Institute of Oncology (ICO), 08907 Badalona, Spain

**Keywords:** metastatic castration-resistant prostate cancer, survivin, YM155, carboplatin

## Abstract

**Background/Objectives**: Metastatic castration-resistant prostate cancer (mCRPC) remains a major clinical challenge due to its aggressive behavior and resistance to therapy. Survivin, a member of the inhibitor of apoptosis protein family, is overexpressed in various cancers and associated with poor prognosis. YM155 (Sepantronium bromide) suppresses survivin expression and has demonstrated antitumor activity in preclinical models. We investigated the association between survivin expression and clinical outcomes in mCRPC patients and evaluated the antitumor activity of YM155, alone and in combination with carboplatin, in mCRPC cell lines. **Methods**: Analysis of publicly available RNA-seq datasets from mCRPC patients was performed to assess correlations between survivin expression and clinical outcomes. Radiographic progression-free survival (rPFS) and overall survival (OS) were estimated using the Kaplan–Meier method and compared via log-rank or Fisher’s exact tests. In vitro assays were conducted on mCRPC cell lines treated with YM155, carboplatin, or both, to evaluate cell viability, clonogenicity, and apoptosis. **Results**: Survivin was significantly overexpressed in mCRPC compared with localized prostate cancer and was even higher in castration-resistant neuroendocrine disease. High survivin levels were associated with shorter OS (*p* = 0.006). In patients treated with platinum-based therapies, high survivin was also linked to shorter rPFS (*p* = 0.01), reduced OS (*p* = 0.006), and a smaller PSA decline (*p* = 0.006). In vitro, YM155 reduced survivin expression, impaired cell viability and colony formation, induced apoptosis, and synergistically enhanced the cytotoxicity of carboplatin. **Conclusions**: Our findings suggest that survivin may serve as a prognostic biomarker and potential therapeutic target in platinum-treated, AR-independent mCRPC. The integration of clinical and functional data provides translational support for combining the survivin inhibitor YM155 with platinum-based therapy. These results warrant further validation in larger patient cohorts and in vivo models.

## 1. Introduction

Prostate cancer (PC) is a leading cause of cancer-related mortality worldwide [1]. Androgen deprivation therapy (ADT) and androgen receptor (AR) signaling inhibitors (ARSIs) are the primary systemic treatment for patients with advanced prostate cancer. While initially effective, nearly all patients with metastatic PC eventually progress to metastatic castration-resistant PC (mCRPC), which is associated with a poor prognosis and a median overall survival (OS) of approximately three years [2,3]. Therapeutic options for mCRPC patients include ARSIs, docetaxel, LuPSMA617 (for those with PSMA-positive disease), radium-223 (for those with bone metastases), PARP inhibitors (for those with BRCA alterations), and cabazitaxel. Platinum-based chemotherapy is sometimes considered, especially in patients with aggressive variant clinical features or pathologic evidence of neuroendocrine prostate cancer (NEPC) [4,5]. Despite multiple available therapies for mCRPC, treatment resistance is common, and there is a clear need for novel approaches to overcome resistance.

A key hallmark of cancer is the ability to evade apoptosis or programmed cell death. This enables tumors to accumulate molecular alterations that drive aggressive biological behavior and lead to treatment resistance [6]. For example, survivin, a key member of the inhibitor of apoptosis protein (IAP) family encoded by the *BIRC5* gene, plays a central role in regulating cell division and inhibiting apoptosis via caspase suppression [7,8,9,10,11,12]. Survivin expression is largely restricted to cancerous tissues, where it has been associated with tumor aggressiveness, higher risk of recurrence, and shorter patient survival, as well as chemotherapy resistance in several cancer types, including PC [7,11,13,14,15,16]. Targeting survivin has been shown to enhance the efficacy of certain cancer treatments, including platinum-based agents like cisplatin and carboplatin [17,18,19].

Cisplatin and carboplatin are among the most widely used chemotherapeutic agents and are known for their ability to induce DNA damage and trigger apoptosis [20]. In mCRPC, carboplatin—either alone or in combination with cabazitaxel—has demonstrated clinical activity in patients with aggressive variant PC (AVPC) and NEPC [21]. AVPC and NEPC are characterized by clinical and pathologic features associated with AR-independence. Platinum-based chemotherapy has also shown activity in DNA repair-deficient mCRPC [22]. The rationale for combining survivin inhibitors with platinum chemotherapy lies in the potential to enhance cell death while overcoming resistance mechanisms.

Several experimental strategies have been developed to target surviving [8], including YM155 (Sepantronium bromide), a small-molecule inhibitor of survivin expression (Appendix A). YM155 is a cell-permeable imidazolium derivative that suppresses *BIRC5* transcription by directly binding to its promoter region and disrupting transcriptional activation [23,24]. Recent studies have revealed that YM155 induces DNA damage and activates cellular stress pathways, including modulation of PI3K/AKT and mTOR signaling. Moreover, YM155 localizes to mitochondria, where it promotes mitochondrial dysfunction, increases glycolytic intermediates, and decreases oxidative phosphorylation and tricarboxylic acid (TCA) cycle activity, ultimately triggering AMP-activated kinase (AMPK) activation [25]. It has demonstrated potent antitumor and antiproliferative activity in various cancer cell lines and mouse models, including CRPC xenografts [26,27,28]. Moreover, YM155 enhanced the efficacy of docetaxel [29], cabazitaxel [30], and platinum compounds [17] in both in vitro and in vivo studies. Phase I and II clinical trials have also confirmed the safety and tolerability of YM155 in patients with advanced, treatment-refractory cancers [31].

Although previous studies have explored the inhibition of survivin in PC, most have focused on restoring sensitivity to taxane-based therapies, such as docetaxel or cabazitaxel [29,30]. Recent bioinformatic analyses, including the work by Yu et al. (2025), have also identified members of the BIRC family—particularly *BIRC5* and *BIRC7*—as potential prognostic and therapeutic biomarkers in PC [32]. However, the potential role of survivin expression in shaping response to platinum-based chemotherapy has not been systematically evaluated in mCRPC. Considering that AR-independent and neuroendocrine-like prostate cancers are more likely to receive platinum treatment, our study specifically addresses the clinical and biological relevance of *BIRC5*/survivin in this disease context.

Based on the association between survivin expression and platinum resistance in other cancer types [11], as well as on the clinical activity of platinum chemotherapy in some patients with AR-independent mCRPC [4], we explored the association between survivin expression and clinical outcomes in mCRPC patients and evaluated the efficacy of YM155—both as monotherapy and in combination with carboplatin—in AR-negative mCRPC cell lines.

## 2. Results

### 2.1. Survivin Is Overexpressed in mCRPC and NEPC and Correlates with Shorter OS

RNA-Seq analysis of survivin (*BIRC5*) mRNA expression levels in datasets of benign prostate, localized PC, CRPC, and NEPC tissue samples [33,34] showed no significant difference between benign tissue and PC. However, survivin was upregulated in mCRPC compared to benign tissue and localized PC, and NEPC samples had the highest levels of survivin expression (Figure 1A,B). Survivin expression was significantly higher in NEPC than in mCRPC (*p* < 0.01), highlighting a potential role of survivin in AR-negative disease and the aggressive neuroendocrine phenotype (Figure 1B). In agreement, we observed a positive association between survivin (*BIRC5*) levels and neuroendocrine markers, including *SYP* (r = 0.654, *p* = 2.81 × 10^−20^), *CHGA* (r = 0.601, *p* = 1.34 × 10^−16^), and *ASCL1* (r = 0.592, *p* = 4.81 × 10^−16^), suggesting a link between survivin upregulation and neuroendocrine differentiation. Conversely, *BIRC5* expression was negatively correlated with *KLK3* expression (r = −0.461, *p* = 1.35 × 10^−9^), further supporting its inverse relationship with AR signaling (Figure 1C). When looking at prostate cancer preclinical models, AR-negative and AR-low cell lines had higher survivin expression compared with AR-positive lines. (Figure 1D). Moreover, survivin (*BIRC5*) expression was significantly higher in CRPC compared to castration-sensitive disease (*p* = 0.021) (Figure 1E). Consistent with these results, survivin (*BIRC5*) expression varied significantly across prostate cancer subtypes (ANOVA *p* = 0.049) among previously profiled preclinical models [35] by ATAC-seq, with higher levels observed in AR-low/AR-negative subtypes, including small-cell-like (SCL), WNT-driven, and NEPC tumors, compared to AR-positive tumors (AR^+^) (Figure 1F). In further support of this, there was an inverse correlation of survivin (*BIRC5*) levels and *AR* and AR target genes such as *KLK2* and *KLK3* across mCRPC (Figure 1G). These findings suggest a progressive increase in survivin expression as PC advances, with a particular impact in AR-independent CRPC.

For the subset of patients with survival data available (*n* = 35), we found that survivin expression levels impacted OS calculated from the diagnosis of metastasis. OS for patients with low survivin expression was 36.7 months, compared to 14 months for those with high expression (HR 2.40, 95% CI 1.15–5.02, *p* = 0.008) (Figure 1H). These differences indicate a potential prognostic value for survivin expression in metastatic PC, with elevated expression correlating with shorter OS.

### 2.2. Survivin Overexpression Is Associated with Response and Outcomes in Platinum-Treated Patients

Given that platinum chemotherapy is sometimes used in mCRPC with AVPC features, including NEPC, we evaluated the clinical associations of survivin (*BIRC5*) expression levels in platinum-treated patients (*n* = 23) [36]. Patients with high survivin expression treated with platinum exhibited significantly shorter OS (8.0 vs. 22.0 months, HR 2.83, 95% CI 1.08–7.37, *p* = 0.006) and rPFS (6.5 vs. 11.1 months, HR 1.71, 95% CI 0.71–4.14, *p* = 0.011) than those with low survivin expression (Figure 2A,B). PSA response analysis showed a greater reduction in PSA levels in patients with low survivin expression compared to those with high survivin expression (*p* = 0.006) (Figure 2C). These results suggest that high survivin expression may be associated with poorer treatment response in mCRPC patients receiving carboplatin.

### 2.3. YM155 Inhibition of Survivin Decreases Cell Proliferation and Induces Cell Death in mCRPC Cell Lines

Western blot analysis showed a marked dose-dependent reduction in survivin protein expression after 72 h of YM155 treatment in the AR-negative mCRPC cell lines DU145 and PC3, supporting the ability of YM155 to target survivin (Figure 3A). MTT assays performed after 72 h of treatment with YM155 doses ranging from 0 to 20 nM revealed a dose-dependent decrease in cell viability, with IC50 values of 8.3 nM for DU145 cells and 3.3 nM for PC3 cells (Figure 3B). Colony formation assays demonstrated a significant reduction in the number of colonies following 72 h of YM155 treatment in a dose-dependent manner in both DU145 and PC3 cells (Figure 3C). Finally, cell death percentages were evaluated in DU145 and PC3 cells after 144 h of treatment with YM155 at 8 nM and 3 nM, respectively. The results showed a significant increase in cell death in both cell lines, further supporting the effect of YM155 (Figure 3D). These results demonstrate that YM155 effectively suppresses survivin expression, reduces cell viability, impairs clonogenic potential, and induces cell death in mCRPC cells.

### 2.4. YM155 Inhibition of Survivin Synergistically Sensitizes mCRPC Cells to Carboplatin Treatment

The combination of YM155 and carboplatin exhibited synergistic effects by reducing cell viability and clonogenic potential, while also inducing cell death in both DU145 and PC3 cell lines. MTT assays conducted after 72 h of treatment demonstrated a dose-dependent reduction in cell viability when the drugs were combined, with a greater effect than either drug alone (Figure 4A,C). The combination index (CI) analysis confirmed the synergistic interaction between YM155 and carboplatin, with CI values < 1, particularly in the PC3 cells (Figure 4B,D). Colony formation assays after 24 h of treatment with the combination of both drugs revealed a significant decrease in the number of colonies compared to that achieved with either drug as a single agent in both cell lines (Figure 4E,F). Additionally, the YM155-plus-carboplatin combination induced significantly higher levels of cell death after 72 h compared to YM155 or carboplatin, whereas in PC3 cells, no statistically significant difference was observed between carboplatin alone and the combination treatment (Figure 4G). These results suggest the potential of YM155-plus-carboplatin as a therapeutic strategy for a subset of advanced PC.

## 3. Discussion

Due to its association with aggressive PC subtypes and poor prognosis [16], survivin has emerged as a potential target in mCRPC. In the present study, we explored the relationship between survivin expression and clinical outcomes and evaluated the possibility of inhibiting survivin with YM155. We confirmed that survivin is overexpressed in aggressive PC subtypes, particularly in AR-low and NEPC tumors, where it is associated with shorter OS. In addition, our in vitro findings strongly indicate that the pharmacological inhibition of survivin using YM155 effectively suppresses its expression.

In our clinical analyses, high survivin (*BIRC5*) expression in mCRPC patients treated with platinum-based therapies was associated with poor outcomes. In addition, patients with lower survivin expression experienced greater reductions in PSA levels and better clinical responses, indicating a potential role for survivin as both a prognostic biomarker and a therapeutic target in this patient population. These findings are in line with previous studies that have linked survivin expression to increased tumor aggressiveness, resistance to apoptosis and DNA-damaging agents, and poor prognosis across various cancer types [8,13,14,15,16,17,18,19].

Our in vitro findings in mCRPC cell lines indicate that YM155 can suppress survivin expression and is associated with decreased cell viability, impaired clonogenic potential, and increased apoptosis. YM155 demonstrated potent cytotoxic effects in both DU145 and PC3 cells, with low IC50 values, highlighting its therapeutic potential. These results align with prior reports demonstrating its efficacy in preclinical cancer models [26,27,28] and its ability to enhance the effectiveness of several drugs used in the treatment of mCRPC [17,29,30].

Importantly, the combination of YM155 with carboplatin exhibited a synergistic effect in DU145 and PC3 AR-negative mCRPC cell lines, as evidenced by reduced cell viability, decreased colony formation, and elevated cell death rates compared to results achieved with monotherapy. This synergism between YM155 and carboplatin suggests that inhibiting survivin may enhance the sensitivity of mCRPC to platinum-based therapies, which have so far shown only transient efficacy in this disease [4]. Notably, in PC3 cells, the combination did not significantly increase apoptosis compared with carboplatin alone; however, Chou–Talalay analysis confirmed synergistic antiproliferative activity (CI < 1). These findings may suggest that the observed synergy in PC3 results primarily from cell-cycle arrest rather than apoptosis induction, consistent with the mitotic role of survivin [7]. Nevertheless, further studies would be required to confirm this hypothesis. Our findings thus support the rationale for combining YM155 with carboplatin as a potential therapeutic strategy to overcome treatment resistance and improve clinical outcomes, particularly in NEPC, where survivin overexpression is more pronounced.

While YM155 was originally described as a transcriptional repressor of *BIRC5* through direct interaction with its promoter, subsequent studies have indicated that its mechanism of action is more complex and not fully elucidated. There is evidence suggesting that YM155 may act indirectly through pathways involving TP53 activation, induction of DNA damage, or disruption of mitochondrial function [37,38]. In line with these findings, Wang et al. (2011) reported that YM155 triggers autophagy-dependent apoptosis in prostate cancer cells [39], while Danielpour et al. (2019) demonstrated that it suppresses mTORC1 activity via AMPK activation, reinforcing its role as a broader stress-response modulator beyond survivin inhibition [40].

Clinically, Tolcher et al. (2012) conducted a phase II clinical trial in taxane-pretreated CRPC patients showing modest single-agent activity but acceptable tolerability [41], and Aoyama et al. (2013) described its pharmacokinetic profile, supporting further exploration in combination strategies [42].

Building on this evidence, our study provides new functional and clinical data in a distinct context—platinum-treated, AR-negative mCRPC—showing that YM155 synergistically enhances carboplatin efficacy. This synergy suggests that survivin inhibition may sensitize resistant prostate cancer cells to DNA-damaging agents, offering a rationale for the biomarker-driven evaluation of this combination.

Whereas previous reports mainly investigated YM155 in combination with taxanes to overcome chemoresistance in PC models [30] or identified survivin family members such as *BIRC5* and *BIRC7* as prognostic biomarkers through bioinformatic analyses [32], our work extends these findings by linking survivin expression to platinum response and by functionally validating the YM155–carboplatin synergy in AR-negative mCRPC models.

On the other hand, while survivin was previously identified as a relevant target in advanced PCa, our work provides a more comprehensive and clinically integrated perspective by correlating survivin expression with real-world treatment responses to platinum in mCRPC patients. This connection has not been fully characterized in earlier studies, particularly in the context of NEPC and AR-low tumors, where therapeutic options remain limited. Moreover, although YM155 was developed more than a decade ago, renewed interest has emerged due to its favorable pharmacological profile and synergy with current treatment regimens, as evidenced in our results. Thus, our study revives and contextualizes survivin as a clinically actionable vulnerability in treatment-resistant disease subtypes.

Our study has certain limitations, including the relatively small number of patients included in the clinical analyses, which precludes generalizing our findings and drawing definite conclusions. This limitation mainly reflects the clinical and technical challenges of studying platinum-treated mCRPC. Platinum chemotherapy is reserved for a small subgroup of patients with AVPC or NEPC, as recommended by the NCCN Guidelines (Version 1.2021). In addition, obtaining adequate tumour tissue is difficult in this setting—over 70% of patients develop bone metastases, and biopsies of osteoblastic lesions often yield limited material or degraded nucleic acids after decalcification. Lymph-node biopsies, though more feasible, are not always accessible. These factors restrict the availability of evaluable samples and explain the limited cohort size, highlighting the practical barriers to molecular studies in this rare but clinically important population.

In addition, we acknowledge the lack of in vivo validation as a limitation of this study. However, the consistency between clinical associations in platinum-treated patients and the in vitro synergistic effects observed in AR-negative models provides strong translational support for our conclusions. Future studies will incorporate in vivo models to confirm the therapeutic potential and safety of the YM155–carboplatin combination.

Nevertheless, our results indicate an association between survivin expression and clinical outcomes that warrants further investigation in larger cohorts. In addition, our study did not include in vivo experiments, which are essential to validate the findings observed in vitro and to reliably assess the safety and efficacy of the YM155-plus-carboplatin combination. Future studies should incorporate in vivo models and expand to other models of CRPC, including NEPC, to strengthen the translational value of our research and provide additional insights into the clinical applicability of survivin-targeted therapies.

In summary, our findings suggest that survivin may serve as a prognostic biomarker and potential therapeutic target in platinum-treated, AR-independent mCRPC. The preclinical data demonstrate that YM155 effectively suppresses survivin expression and enhances the cytotoxic activity of carboplatin, providing translational support for this combination strategy. By integrating molecular profiling, clinical outcomes, and functional assays, this work highlights survivin as a biologically relevant target in mCRPC. Given the exploratory and hypothesis-generating scope of this work, further validation in larger patient cohorts and in vivo models will be essential to confirm these findings and define their clinical relevance.

## 4. Materials and Methods

### 4.1. Computational Analysis of Human PC Data

We analyzed previously published RNA-Seq datasets generated by the group of Dr. Himisha Beltran and collaborators to assess mRNA expression levels for *BIRC5*, neuroendocrine genes, and AR/luminal markers across prostate cancer subtypes (34 benign, 68 localized PC, 31 CRPC, and 22 NEPC) [33,34,36]. Sequencing reads were aligned to the human reference genome (hg19/GRCh37) using STAR software (v2.3.0e; Cold Spring Harbor Laboratory, Cold Spring Harbor, NY, USA). For each sample, the High-Throughput Sequencing framework (HTSeq) v0.6.1 (Python framework developed at EMBL, Heidelberg, Germany) was employed to generate read counts, while Cufflinks (v2.2.1; Trapnell Lab, University of Washington, Seattle, WA, USA) was used to calculate FPKM (Fragments Per Kilobase of transcript per Million mapped reads) values. In addition, we analyzed RNA-Seq data from previously published preclinical models [35] and clinical samples [43].

### 4.2. Cell Lines

DU145 (HTB-81™) and PC3 (CRL-1435™) mCRPC cell lines (American Type Culture Collection; ATCC, Manassas, VA, USA) were used in this study. DU145 and PC3 cell lines were selected as representative models of androgen-insensitive metastatic prostate cancer, widely recognized as models of mCRPC [44]. Briefly, DU145 cells were maintained in RPMI medium (11875093, Thermo Fisher Scientific, Waltham, MA, USA), while PC3 cells were grown in Ham’s F12K medium (21127022, Thermo Fisher Scientific, Waltham, MA, USA), both supplemented with 10% heat-inactivated fetal bovine serum (FBS, FBS-12A, Reactiva, Barcelona, Spain) and 1% penicillin–streptomycin (15140-122, Thermo Fisher Scientific, Waltham, MA, USA). The cell cultures were incubated at 37 °C in a 5% CO_2_ environment, regularly checked for mycoplasma contamination, and verified through short tandem repeat profiling.

### 4.3. Drugs

YM155 (Sepantronium bromide; HY-10050, MedChemExpress, Monmouth Junction, NJ, USA) and carboplatin (HY-17393, MedChemExpress, Monmouth Junction, NJ, USA ) were prepared in dimethyl sulfoxide (DMSO, D8418, Sigma-Aldrich, St. Louis, MO, USA) at a concentration of 10 mM and stored at 4 °C and RT, respectively. Further dilutions of each drug were made in culture medium to the final concentrations before use.

### 4.4. MTT Cell Viability Assay

Drug cytotoxicity was assessed by the 3-(4,5-dimethylthiazol-2-yl) 2,5-diphenyltetrazolium bromide (MTT) assay (M6494, Thermo Fisher Scientific, Waltham, MA, USA). DU145 and PC3 cells were seeded in 96-well microtiter plates (Thermo Fisher Scientific, Waltham, MA, USA) at 6000 or 8000 cells/well, respectively, and allowed to attach. Medium containing different drug concentrations of YM155, carboplatin, and their combination was added after 24 h. After 72 h of treatment, MTT was added absorbance was measured at 570 nm using a Varioskan Flash microplate reader (Thermo Scientific, Waltham, MA, USA). Doses for 10–90% cell viability were determined by the median-effect line method.

### 4.5. Cytotoxicity of Drug Combinations

Cell viability for drug combination studies was determined using the MTT assay described in Section 4.4, and drug interactions were analyzed according to the Chou–Talalay method. The cytotoxic effects of the drug combinations were evaluated using serial dilutions of both drugs at concentrations equivalent to 1/8, 1/4, 1/2, 1, 2, and 4 times their respective IC50 values, as assessed by the MTT assay. Fractional survival was determined by dividing the number of viable cells in drug-treated wells by those in untreated control wells. The synergistic interaction between the treatments was assessed by calculating the Combination Index (CI) using Compusyn Software v1.0 (ComboSyn Inc., Paramus, NJ, USA), following the Chou–Talalay method, as previously reported [45]. In this approach, a CI value < 1 indicates synergy, a CI > 1 indicates antagonism, and CI = 1 indicates an additive effect.

### 4.6. Colony Formation Assay

To further evaluate the cytotoxicity of YM155 and carboplatin, both alone and in combination, colony formation assays were performed as previously described [45]. Briefly, a serial dilution of DU145 and PC3 cells was made in order to seed 500 cells/well in a six-well plate (Thermo Fisher Scientific), and cells were left for 24 h to adhere. Different dilutions of the drugs were then added for 24 or 72 h. For monotherapy experiments, YM155 was tested at concentrations ranging from 2.5 to 40 nM for DU145 (2.5, 5, 10, 20, and 40 nM) and from 0.5 to 10 nM for PC3 (0.5, 1, 2.5, 5, and 10 nM). For drug combination experiments, the following concentrations were used: DU145—carboplatin 0.5 µM and YM155 10 nM; PC3—carboplatin 2.5 µM and YM155 2.5 nM. Cells were left a total of 10 days in culture for colonies to form, and regular medium changes were performed. Cells were subsequently washed with PBS, fixed with a methanol/acetic acid (3:1) solution for 10 min, and stained with a 0.5% crystal violet solution (C0775, Sigma-Aldrich, St. Louis, MO, USA) for 10 min. After staining, cells were washed with PBS, and colonies were counted with ImageJ software v1.54 (National Institutes of Health, Bethesda, MD, USA).

### 4.7. Western Blot Assay

Western blotting was carried out as previously described [45]. Briefly, cells were washed with PBS and lysed using a radioimmunoprecipitation assay (RIPA+) buffer. Protein concentrations were determined using the DC Protein Assay (5000111, Bio-Rad Laboratories, Richmond, CA, USA). For the analysis, 50 µg of protein from each sample were separated on a 10% sodium dodecyl sulfate–polyacrylamide gel (SDS-PAGE) (NP0301BOX, Thermo Fisher Scientific, Waltham, MA, USA) via electrophoresis and transferred onto a polyvinylidene difluoride (PVDF) membrane (1620177, Bio-Rad Laboratories, Richmond, CA, USA) using wet transfer. Membranes were blocked with Odyssey blocking buffer (927-40000, LI-COR Biosciences, Lincoln, NE, USA) for 2 h and subsequently incubated overnight at 4 °C with specific primary antibodies against Survivin (Ref. EP2880Y; 1:1000, Abcam, Cambridge, MA, USA) and β-actin ( #T6074; 1:2000, Sigma-Aldrich, St. Louis, MO, USA). Following primary incubation, membranes were treated with IRDye rabbit and mouse secondary antibodies (1:10,000) (RDye 800CW/680RD, LI-COR Biosciences, Lincoln, NE, USA), scanned, and analyzed using the LiCor Odyssey 9120 infrared imaging system (LI-COR Biosciences, Lincoln, NE, USA). Protein bands were normalized to the corresponding β-actin band from the same sample. Uncropped Western blot images are provided in Appendix A.

### 4.8. Cell Death Assay

Annexin V (Alexa Fluor^®^ 647 Annexin V, 640912, BioLegend, San Diego, CA, USA) and DAPI (62248, Thermo Fisher Scientific, Waltham, MA, USA) were used and analyzed on a flow cytometry Fortessa 4 HTS instrument (BD LSRFortessa™ 4 HTS, BD Biosciences, San Jose, CA, USA) as previously described [46]. Viable cells were defined as negative for Annexin-V and DAPI, and cell death was expressed as 100% viable cells. Analyses were performed using FlowJo software v10.9 (BD FlowJo LLC, Ashland, OR, USA).

### 4.9. Statistical Analyses

In all in vitro functional assays, data are presented as mean ± SEM of at least three independent biological replicates with three internal technical replicates, and the statistical analysis was performed with Graphpad Prism V.4 software (GraphPad Software Inc., San Diego, CA, USA). Statistical differences between IC50 were determined by graphic representation of dose–response curves and subsequent non-linear regression analysis and F-test. For colony formation and apoptosis assays, *p*-values were calculated using a two-tailed Student’s *t*-test. The null hypothesis was that no differences existed between the experimental and control groups compared in each independent analysis. Significance was set at *p* ≤ 0.05.

A decline in prostate-specific antigen (PSA) was evaluated according to Prostate Cancer Clinical Trials Working Group 3 (PCWG3) guidelines (PSA decline > 50%) at the time of radiographic progression. Radiographic progression-free survival (rPFS) and OS were estimated by the Kaplan–Meier method and compared using the log-rank or Fisher’s exact test. PSA decline was evaluated using Fisher’s exact test. Significance was set at 5% association with PSA decline. Samples with *BIRC5* expression Z-scores above zero were classified as high, and the rest as low.

*BIRC5* expression differences across groups and disease states were assessed using one-way ANOVA and Welch’s *t*-test, respectively. Pearson correlation analysis was performed to identify associations between *BIRC5* expression and other genes, reporting correlation coefficients (r) and significance levels (*p*-values). Box plots were generated to visualize expression distributions, with statistical significance annotated. A *p*-value of < 0.05 was considered statistically significant.

## Figures and Tables

**Figure 1 pharmaceuticals-18-01752-f001:**
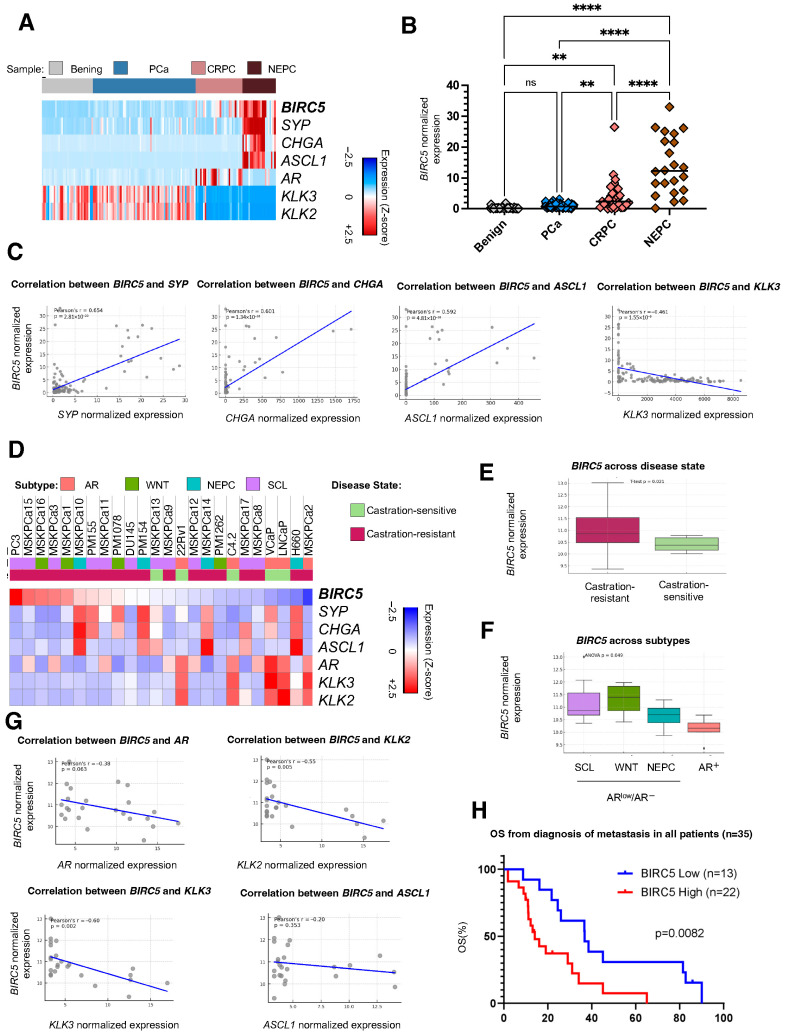
Survivin (BIRC5) expression and its association with overall survival in subsets of metastatic prostate cancer: (**A**) A heatmap of survivin gene (*BIRC5*), adeno and neuroendocrine (NE) markers expression in benign, PC, castration-resistant PC (CRPC), and neuroendocrine PC (NEPC) tissue samples. (**B**) Survivin (*BIRC5*) expression across PC stages. RNA-seq analysis shows the distribution of *BIRC5* mRNA expression (RPKM) in benign, prostate cancer (PC), castration-resistant prostate cancer (CRPC), and neuroendocrine prostate cancer (NEPC) tissue samples. Statistical significance was assessed by one-way ANOVA followed by Welch’s *t*-test; ** *p* < 0.01, **** *p* < 0.0001, and ns (not significant) refer to differences between groups. (**C**) Correlation between BIRC5 expression and the expression of NE markers, including *SYP*, *CHGA*, and *ASCL1*, and Adeno marker *KLK3*. Each scatter plot represents the relationship between *BIRC5* expression (x-axis) and the expression levels of *SYP*, *CHGA*, *ASCL1*, and *KLK3* (from left to right sub-panel). Pearson’s correlation coefficient (r) and corresponding *p*-values are shown in each plot. (**D**) Heatmap representation of gene expression data, displaying variations across different PC cell lines and organoids from Gene Expression Omnibus (GSE199190). The heatmap shows a color gradient from blue (low expression) to red (high expression) of *BIRC5* and is annotated with disease state and ATAC-seq associated subtypes, including AR, WNT, NEPC, and stem cell-like (SCL) subtypes. (**E**) The expression of *BIRC5* across disease states; the unpaired *t*-test was used for comparing between two groups. (**F**) The expression of *BIRC5* across ATAC-seq associated subtypes; data were analyzed by one-way analysis of variance (ANOVA) followed by Tukey’s multiple comparison tests. (**G**) Correlation between *BIRC5* expression and the expression of adeno markers, including *AR*, *KLK2*, and *KLK3*, and NE marker *ASCL1*. Each scatter plot represents the relationship between *BIRC5* expression (x-axis) and the expression levels of *AR*, *KLK2*, *KLK3*, and *ASCL1* (from top left to right bottom sub-panel). Pearson’s correlation coefficient (r) and corresponding *p*-values are shown in each plot. (**H**) Overall survival (OS) from the diagnosis of metastasis according to *BIRC5* expression levels. Blue lines represent patients with low expression (z-score < 0), and red lines represent patients with high expression (z-score >0). Log-rank test *p*-values refer to survival differences between groups.

**Figure 2 pharmaceuticals-18-01752-f002:**
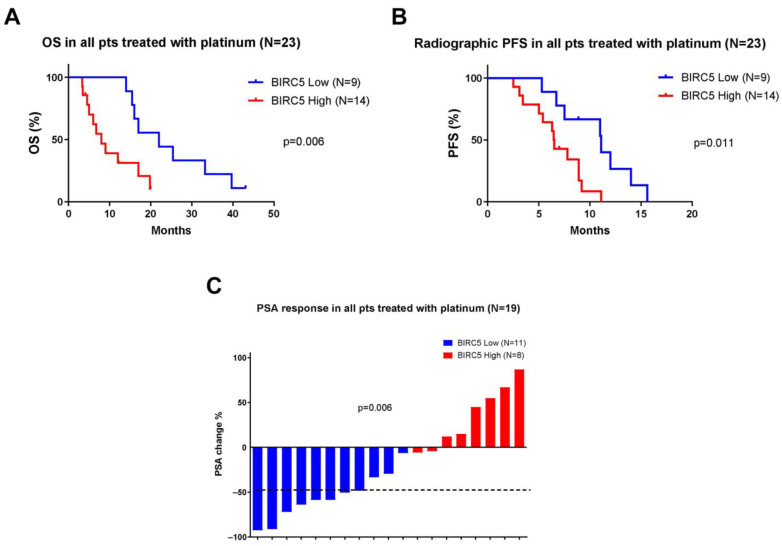
Correlation between survivin (*BIRC5*) expression and clinical outcomes in platinum-treated metastatic castration-resistant prostate cancer (mCRPC) patients: (**A**,**B**) Kaplan–Meier curves showing overall survival (OS) (**A**) and radiographic progression-free survival (rPFS). (**B**) Results according to *BIRC5* expression levels. Blue lines represent patients with low expression, and red lines represent patients with high expression. (**C**) Waterfall plots depicting changes in PSA levels (%) according to *BIRC5* expression levels (low [blue] vs. high [red]). Negative values represent prostate-specific antigen (PSA) decline. Survival analyses were performed using the Kaplan–Meier method and compared using the log-rank test.

**Figure 3 pharmaceuticals-18-01752-f003:**
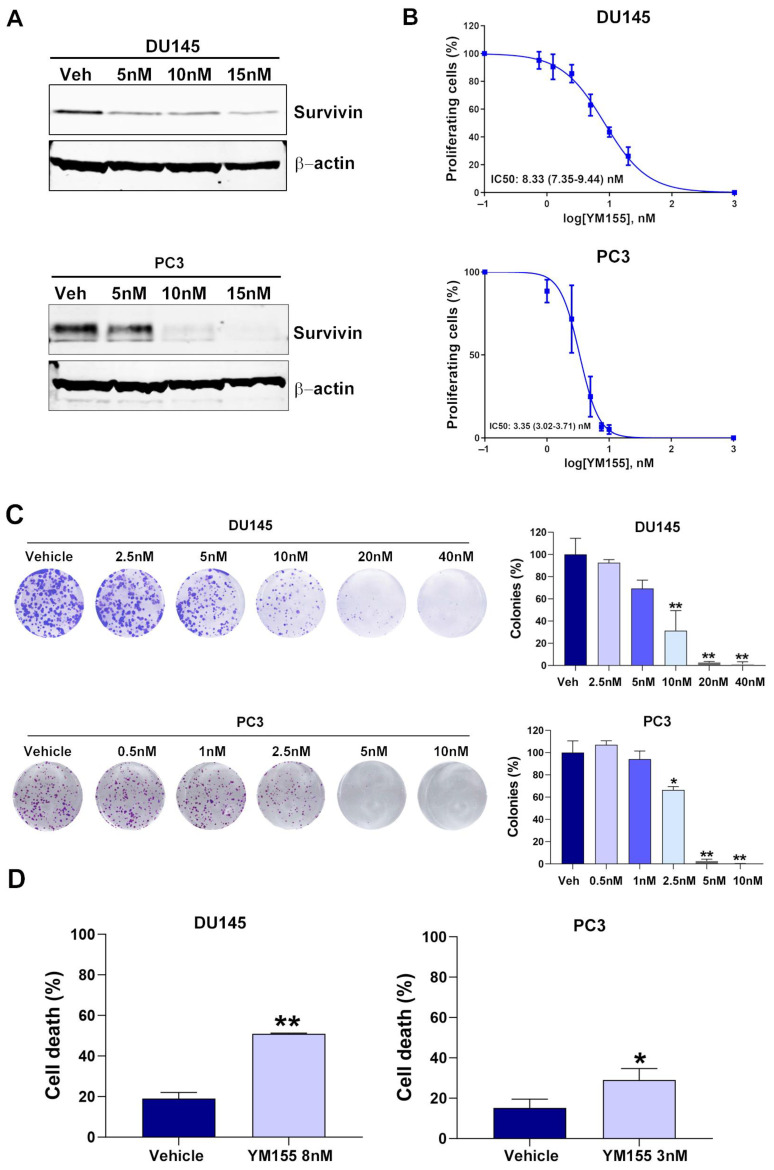
Effects of YM155 on survivin expression, cell viability, colony formation, and cell death in metastatic castration-resistant prostate cancer cell lines: (**A**) Western blot analysis of survivin protein expression in DU145 and PC3 cells treated with increasing doses of YM155 for 72 h. Beta-actin was used as an endogenous control. (**B**) Dose–response curve for DU145 and PC3 cells after YM155 treatment at 0–20 nM for 72 h (mean ± SEM). IC50 value is shown as mean (95% CI). (**C**) Representative colony assay images (left panel) and bar graph (right panel) showing the percentage (mean ± SEM) of colonies in DU145 and PC3 cells after YM155 treatment for 72 h at the indicated doses. * *p* < 0.05 and ** *p* < 0.01 relative to vehicle. (**D**) Bar graph representing the percentage (mean ± SEM) of cell death assessed by annexin V and DAPI staining after treatment with YM155 for 144 h at the indicated doses in DU145 (left panel) and PC3 (right panel) cells. * *p* < 0.05 and ** *p* < 0.01 relative to vehicle. Data represent mean ± SEM of three independent biological experiments (*n* = 3), each performed in technical triplicate. Veh: vehicle. IC_50_, half-maximal inhibitory concentration. Statistical significance between treatment groups was determined using a two-tailed Student’s *t*-test.

**Figure 4 pharmaceuticals-18-01752-f004:**
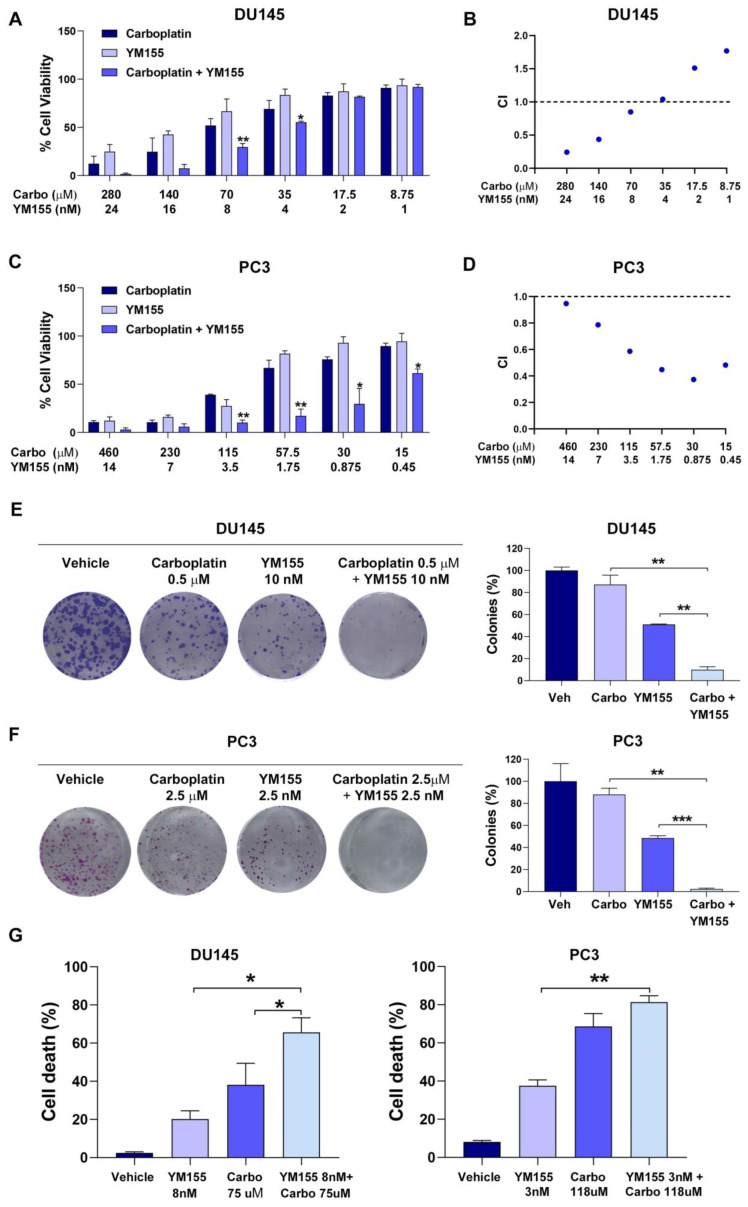
Synergistic effects of YM155 and carboplatin on cell viability, clonogenic potential, and cell death in metastatic castration-resistant prostate cancer cell lines: (**A**) Bar graph representing the percentage (mean ± SEM) of cell viability after treatment with YM155, carboplatin, or the combination of both drugs for 72 h at the indicated doses in DU145 cells. * *p* < 0.05 and ** *p* < 0.01 indicate differences between treatments. (**B**) Dot plot representing combination index (CI) values calculated for each dose of the combination treatment in DU145 cells. (**C**) Bar graph representing the percentage (mean ± SEM) of cell viability after treatment with YM155, carboplatin, or the combination of both drugs for 72 h at the indicated doses in PC3 cells. * *p* < 0.05 and ** *p* < 0.01 indicate differences between treatments. (**D**) Dot plot representing combination index (CI) values calculated for each dose of the combination treatment in PC3 cells. (**E**,**F**) Representative colony assay images (left panel) and bar graph (right panel) representing the percentage (mean ± SEM) of colonies in (**E**) DU145 and (**F**) PC3 cells after treatment with YM155, carboplatin, or the combination of both drugs for 24 h at the indicated doses. ** *p* < 0.01 and *** *p* < 0.001 indicate differences between treatments. (**G**) Bar graph representing the percentage (mean ± SEM) of cell death assessed by annexin V and DAPI staining after treatment with YM155, carboplatin, or the combination of both drugs for 72 h at the indicated doses in DU145 (left panel) and PC3 (right panel) cells. * *p* < 0.05 and ** *p* < 0.01 indicate differences between treatments. Data represent mean ± SEM of three independent biological experiments (*n* = 3), each performed in technical triplicate. Carbo: carboplatin; Veh: vehicle. Statistical significance between treatment groups was determined using a two-tailed Student’s *t*-test.

## Data Availability

The original contributions presented in this study are included in the article/Appendix A. Further inquiries can be directed to the corresponding author.

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
