# Peer review of "YM155 Inhibition of Survivin Enhances Carboplatin Efficacy in Metastatic Castration-Resistant Prostate Cancer"

_pharmaceuticals, 2025, doi:10.3390/ph18111752_

Round 1
Reviewer 1 Report (Previous Reviewer 1)
Comments and Suggestions for Authors
In this study, the authors investigated the association between survivin expression and clinical outcomes in metastatic castration-resistant prostate cancer (mCRPC) patients and evaluated the efficacy of YM155 – both as monotherapy and in combination with carboplatin – in AR-negative mCRPC cell lines. The authors suggested that survivin may serve as both a therapeutic target and a prognostic biomarker in mCRPC. YM155 may help overcome resistance to platinum-based chemotherapy, warranting further investigation of this combination strategy.
Comments:
This is a resubmitted manuscript. The authors have carefully responded the reviewer’s comments and the revised manuscript has a great improvement.
There are only some minor comments as follows:
(1) In the Results of this revised manuscript, lines 240-242, the description needs to be revised because Figure 4G for PC3 cells showed no significant difference between carboplatin alone and YM155+carboplatin.
(2) The authors have reduced the numbers of self-citations. The references cited in this manuscript appropriate and relevant to this research.
Author Response
Please see the attachment

Reviewer 2 Report (Previous Reviewer 2)
Comments and Suggestions for Authors
The authors addressed most of my previous concerns, which I reported in my earlier review, and I am thankful to them; however, I still insist on including the chemical structure of the compound within the article, not as supplementary material.
I find the paper better than before.
Author Response
Please see the attachment

Reviewer 3 Report (Previous Reviewer 3)
Comments and Suggestions for Authors
I have completed the second review of the manuscript entitled ‘YM155 Inhibition of Survivin Enhances Carboplatin Efficacy in Metastatic Castration-Resistant Prostate Cancer’. After carefully analysing your responses to the observations and comments made in the first evaluation, I consider that you have addressed each of the points raised accurately and satisfactorily. The corrections made to the abstract, as well as to the materials and methods, results, and discussion sections, have strengthened the clarity and scientific rigour of the work. In light of these improvements, I recommend that the article be accepted for publication.
Author Response
Please see the attachment

This manuscript is a resubmission of an earlier submission. The following is a list of the peer review reports and author responses from that submission.
Round 1
Reviewer 1 Report
Comments and Suggestions for Authors
In this study, the authors investigated the association between survivin expression and clinical outcomes in metastatic castration-resistant prostate cancer (mCRPC) patients and evaluated the efficacy of YM155 – both as monotherapy and in combination with carboplatin – in AR-negative mCRPC cell lines. The authors suggested that survivin may serve as both a therapeutic target and a prognostic biomarker in mCRPC. YM155 may help overcome resistance to platinum-based chemotherapy, warranting further investigation of this combination strategy.
Comments:
The reviewer has some concerns as follows:
- One of the major concerns is that the novelty of this study. Miyao et al. have shown that inhibition of survivin by YM155 overcomes cabazitaxel resistance in castration-resistant prostate cancer cells in vitro and in vivo (doi: 10.21873/anticanres.14512). Carboplatin is also know to use for metastatic castration-resistant prostate cancer, often in combination with other chemotherapy drugs like docetaxel or cabazitaxel, to improve response rates (e.q. Corn et al., doi: 10.1016/S1470-2045(19)30754-5.). Therefore, the synergistic effect of YM155+ carboplatin on mCRPC is predictable. Moreover, Yu et al. have recently demonstrated the potential of BIRC5 (Survivin) or BIRC7 genes as prognostic biomarkers, offering new insights into possible targets for the development of therapeutic biomarkers and immunotherapeutic for prostate cancer (Yu et al. Exploring BIRC family genes as prognostic biomarkers and therapeutic targets in prostate cancer. Discov Oncol. 2025 Feb 26;16(1):240.). Therefore, the findings of this study are not impressive.
- In Figure 2, the data for correlation between survivin (BIRC5) expression and clinical outcomes in platinum-treated mCRPC patients are not reliable because of the sample size is too small.
- In Figures 3 and 4, the sample size (n number) needs to be added in the legends.
- In Figure 4G for PC3 cells, there is no significant difference between carboplatin alone and YM155+carboplatin, indicating there is no synergistic effect of both in PC3 cells.
- The lack of experimental evidence from animal models makes it impossible to verify the reliability of the conclusions.
- The reference list needs to be modified. There are too many self-citations in this manuscript (13 in total 40 references).
- Overall, in the present state, the results cannot support the conclusions.
Author Response
Dear Reviewer,
We sincerely thank you for your careful reading of our manuscript and your constructive comments. We have revised the manuscript thoroughly in response to your suggestions, which have helped us to significantly improve the clarity, accuracy, and completeness of our review. All modifications in the manuscript have been made using the track changes function. Below we provide a detailed, point-by-point response to each of your comments.
Reviewer Comment 1:
“One of the major concerns is that the novelty of this study. Miyao et al. have shown that inhibition of survivin by YM155 overcomes cabazitaxel resistance in castration-resistant prostate cancer cells in vitro and in vivo (doi: 10.21873/anticanres.14512). Carboplatin is also know to use for metastatic castration-resistant prostate cancer, often in combination with other chemotherapy drugs like docetaxel or cabazitaxel, to improve response rates (e.q. Corn et al., doi: 10.1016/S1470-2045(19)30754-5.). Therefore, the synergistic effect of YM155+ carboplatin on mCRPC is predictable. Moreover, Yu et al. have recently demonstrated the potential of BIRC5 (Survivin) or BIRC7 genes as prognostic biomarkers, offering new insights into possible targets for the development of therapeutic biomarkers and immunotherapeutic for prostate cancer (Yu et al. Exploring BIRC family genes as prognostic biomarkers and therapeutic targets in prostate cancer. Discov Oncol. 2025 Feb 26;16(1):240.). Therefore, the findings of this study are not impressive”.
Authors Response:
We sincerely thank the reviewer for this thoughtful and constructive comment. We fully agree that it is essential to clarify both the original contributions of our study and its therapeutic relevance, as well as to acknowledge and contextualize previous publications on survivin and YM155 in prostate cancer.
We acknowledge that surviving (BIRC5) and YM155 are not conceptually new. The novelty of our work lies in its translational and context-specific evaluation of survivin in androgen receptor (AR)-independent and neuroendocrine-like metastatic castration-resistant prostate cancer (mCRPC), particularly in patients treated with platinum chemotherapy — a clinical setting that, to our knowledge, has not been previously linked to BIRC5 expression or YM155 efficacy.
Specifically, our study provides:
- Clinical evidence showing that high BIRC5 expression correlates with poorer overall and progression-free survival, as well as reduced PSA response, in platinum-treated mCRPC patients.
- Biological positioning of survivin within AR-low/NEPC-like prostate cancer states.
- Functional validation that inhibition of survivin with YM155 synergistically enhances carboplatin efficacy in AR-negative mCRPC cell models.
In response to the reviewer’s suggestion, we have also incorporated the following articles—Wang et al., 2011; Tolcher et al., 2012; Aoyama et al., 2013; and Danielpour et al., 2019—into the Discussion section to provide a more comprehensive contextualization of YM155’s previously reported mechanisms and clinical studies. These references complement our findings and reinforce the novelty of evaluating survivin inhibition and YM155–platinum synergy in AR-independent mCRPC.
Previous studies mainly focused on taxane resistance or non-platinum settings and did not evaluate survivin as a predictive biomarker or assess the combination of YM155 with platinum compounds in AR-negative prostate cancer models (Miyao et al., 2020). None, to our knowledge, have demonstrated a link between BIRC5 expression and treatment outcomes in mCRPC. Our study fills this gap by integrating clinical, molecular, and functional data to show that survivin inhibition enhances platinum efficacy in AR-independent mCRPC, thus providing both mechanistic and translational evidence for this therapeutic approach.
We also acknowledge Yu et al. (2025), who identified BIRC5 and BIRC7 as potential prognostic biomarkers through in silico analyses. However, their study did not include platinum-treated patients or functional assays. Our work extends these observations by demonstrating a direct association between BIRC5 expression and platinum response and by validating survivin inhibition as a sensitizer to carboplatin in AR-negative mCRPC.
In conclusion, our manuscript does not claim the first discovery of survivin or YM155; rather, it provides new clinical–translational evidence that BIRC5 stratifies benefit from platinum in mCRPC and that survivin inhibition synergizes with carboplatin in AR-negative prostate cancer models, thereby motivating biomarker-selected clinical evaluation of the YM155 + platinum strategy in AR-independent/NEPC-like mCRPC.
To further emphasize the novelty and specific contribution of our work, we have introduced the following clarifications in the revised manuscript:
Page 3; Line 104: Although previous studies have explored the inhibition of survivin in PC, most have focused on restoring sensitivity to taxane-based therapies, such as docetaxel or cabazitaxel [29,30]. Recent bioinformatic analyses, including the work by Yu et al. (2025), have also identified members of the BIRC family—particularly BIRC5 and BIRC7—as potential prognostic and therapeutic biomarkers in PC [32]. However, the potential role of survivin expression in shaping response to platinum-based chemotherapy has not been systematically evaluated in mCRPC. Considering that AR-independent and neuroendocrine-like prostate cancers are more likely to receive platinum treatment, our study specifically addresses the clinical and biological relevance of BIRC5/survivin in this disease context.
Page 12; Line 293: In line with these observations, Wang et al. (2011) reported that YM155 triggers autophagy-dependent apoptosis in prostate cancer cells [39], while Danielpour et al. (2019) demonstrated that it suppresses mTORC1 activity via AMPK activation, reinforcing its role as a broader stress-response modulator beyond survivin inhibition [40].
Clinically, Tolcher et al. (2012) conducted a phase II clinical trial in taxane-pretreated CRPC patients showing modest single-agent activity but acceptable tolerability [41], and Aoyama et al. (2013) described its pharmacokinetic pro-file, supporting further exploration in combination strategies [42].
Building on this evidence, our study provides new functional and clinical data in a distinct context—platinum-treated, AR-negative mCRPC—showing that YM155 synergistically enhances carboplatin efficacy. This synergy suggests that survivin inhibition may sensitize resistant prostate cancer cells to DNA-damaging agents, offering a rationale for the biomarker-driven evaluation of this combination.
Page 12; Line 307: Whereas previous reports mainly investigated YM155 in combination with tax-anes to overcome chemoresistance in PC models [30] or identified survivin family members such as BIRC5 and BIRC7 as prognostic biomarkers through bioinformatic analyses [32], our work extends these findings by linking survivin ex-pression to platinum response and by functionally validating the YM155–carboplatin synergy in AR-negative mCRPC models.
Reviewer Comment 2:
In Figure 2, the data for correlation between survivin (BIRC5) expression and clinical outcomes in platinum-treated mCRPC patients are not reliable because of the sample size is too small.
Authors Response:
We thank the reviewer for this valuable comment and fully acknowledge that the sample size of the platinum-treated mCRPC cohort is limited. This limitation reflects the clinical reality that platinum-based chemotherapy is not a standard treatment for all mCRPC patients but is recommended by the NCCN Guidelines (Version 1.2021) only for fit patients with aggressive variant prostate cancer (AVPC) or neuroendocrine differentiation. Consequently, the eligible patient population is small, and assembling large, homogeneous platinum-treated cohorts remains inherently difficult.
Moreover, obtaining adequate tumor tissue samples in this disease setting poses significant logistical and technical challenges. Over 70% of patients with metastatic prostate cancer develop bone metastases, and nearly half present bone-predominant disease. Despite image guidance, biopsies of osteoblastic bone metastases are often technically complex and may yield insufficient tumor tissue, while decalcification procedures can compromise nucleic acid quality and quantity. Lymph-node biopsies, although more successful, are not always feasible, particularly for deep pelvic or retroperitoneal nodes. These factors further limit the availability of evaluable samples for molecular analysis in this population.
Despite these challenges, the associations between high BIRC5 expression and poorer clinical outcomes (OS, rPFS, and PSA response) were statistically significant (p < 0.01) and directionally consistent across endpoints. The results also align with our in vitro data showing that survivin inhibition sensitizes AR-negative cells to carboplatin, reinforcing biological plausibility.
We now explicitly discuss this limitation in the revised Discussion section of the manuscript:
Page 12; Line 325: This limitation mainly reflects the clinical and technical challenges of studying platinum-treated mCRPC. Platinum chemotherapy is reserved for a small subgroup of patients with AVPC or NEPC, as recommended by the NCCN Guidelines (Version 1.2021). In addition, obtaining adequate tumour tissue is difficult in this setting—over 70% of patients develop bone metastases, and biopsies of osteoblastic lesions often yield limited material or degraded nucleic acids after decalcification. Lymph-node biopsies, though more feasible, are not always accessible. These factors restrict the availability of evaluable samples and explain the limited cohort size, highlighting the practical barriers to molecular studies in this rare but clinically important population.
Reviewer Comment 3:
In Figures 3 and 4, the sample size (n number) needs to be added in the legends.
Authors Response:
We thank the reviewer for this observation. The legends of Figures 3 and 4 already indicated that all results were obtained from at least three independent experiments; however, we understand that the reviewer refers to specifying the exact number of replicates (n) used for each quantitative analysis.
Accordingly, we have revised the figure legends to explicitly include the sample size. The updated legends now read:
Figure 3: Data represent mean ± SEM of three independent biological experiments (n = 3), each performed in technical triplicate.
Figure 4: Data represent mean ± SEM of three independent biological experiments (n = 3), each performed in technical triplicate.
Reviewer Comment 4:
In Figure 4G for PC3 cells, there is no significant difference between carboplatin alone and YM155 + carboplatin, indicating there is no synergistic effect of both in PC3 cells.
Authors response:
We thank the reviewer for this comment and appreciate the opportunity to clarify this point. The synergistic interaction between YM155 and carboplatin was evaluated quantitatively using the Chou–Talalay combination index (CI) method based on MTT viability assays (Figures 4A–D). As shown in Figure 4D, all tested combinations in PC3 cells yielded CI < 1, indicating synergy, which was in fact stronger in PC3 cells than in DU145 cells (Figure 4B).
Figure 4G, in contrast, presents the percentage of apoptotic cell death (Annexin V/DAPI) after mono- and combination treatments. We agree with the reviewer that, in PC3 cells, the increase in apoptosis with the combination did not reach statistical significance compared with carboplatin alone. However, this assay measures late-stage apoptosis, whereas the MTT and CI analyses primarily reflect antiproliferative effects. The lack of statistical significance in apoptosis therefore does not contradict the synergistic reduction in cell viability observed in the CI analysis.
Given that survivin is essential for mitosis and cell-cycle regulation, these findings may suggest that the interaction between YM155 and carboplatin in PC3 cells is predominantly antiproliferative, potentially mediated by cell-cycle arrest rather than apoptosis induction. Nevertheless, we acknowledge that additional mechanistic studies would be required to confirm this hypothesis.
This clarification has been included in the revised Discussion section:
Pàg.11; Line 280: Of note, in PC3 cells, the combination did not significantly increase apoptosis com-pared with carboplatin alone; however, Chou–Talalay analysis confirmed synergistic antiproliferative activity (CI < 1). These findings may suggest that the observed synergy in PC3 results primarily from cell-cycle arrest rather than apoptosis induction, consistent with the mitotic role of survivin [7]. Nevertheless, further studies would be required to confirm this hypothesis.
Reviewer Comment 5:
The lack of experimental evidence from animal models makes it impossible to verify the reliability of the conclusions.
Authors response:
We thank the reviewer for this important observation and fully acknowledge the limitation regarding the absence of in vivo experiments. Our study was conceived as a translational investigation aiming to bridge clinical outcomes with mechanistic biology rather than to provide preclinical validation. Specifically, we analyzed a real-world cohort of platinum-treated mCRPC patients, demonstrating that high BIRC5 expression correlates with poorer overall and progression-free survival, and complemented these findings with functional in vitro assays confirming that survivin inhibition enhances carboplatin efficacy in AR-negative models.
While in vivo validation would certainly strengthen our conclusions, the concordance between patient-derived clinical data and experimental results provides consistent biological plausibility supporting our main hypothesis. We have explicitly acknowledged this limitation in the revised Discussion and highlighted that future studies will include in vivo validation to confirm and extend these observations:
Page 13; Line 335: In addition, we acknowledge the lack of in vivo validation as a limitation of this study. However, the consistency between clinical associations in platinum-treated patients and the in vitro synergistic effects observed in AR-negative models provides strong translational support for our conclusions. Future studies will incorporate in vivo models to confirm the therapeutic potential and safety of the YM155 + carboplatin combination.
Reviewer Comment 6:
The reference list needs to be modified. There are too many self-citations in this manuscript (13 in total 40 references).
Authors response:
We thank the reviewer for this thoughtful observation. Following the reviewer’s comment, we carefully re-evaluated all references to ensure that each citation is essential, scientifically justified, and directly relevant to the content of this study.
Some of the self-citations originally included corresponded to our group’s previous peer-reviewed publications, which provided necessary methodological and conceptual background—particularly in relation to platinum response and AR-independent prostate cancer biology. Nonetheless, we fully acknowledge the reviewer’s concern and have revised the reference list to reduce the number of self-citations and strengthen the diversity of the cited literature.
Specifically, we have removed or replaced citations that were not strictly required for contextual or data support, as detailed below:
Replaced references:
Davies, A.; Conteduca, V.; Zoubeidi, A.; Beltran, H. (2019). Biological Evolution of Castration-Resistant Prostate Cancer. Eur Urol Focus, 5(2), 147–154; replaced by Ceder, Y.; Bjartell, A.; Culig, Z.; Rubin, M. A.; Tomlins, S.; Visakorpi, T. (2016). The Molecular Evolution of Castration-Resistant Prostate Cancer. Eur Urol Focus, 2(5), 506–513.
Yamada, Y.; Beltran, H. (2021). The Treatment Landscape of Metastatic Prostate Cancer. Cancer Lett, 519, 20–29; replaced by Posdzich, P.; Darr, C.; Hilser, T.; Wahl, M.; Herrmann, K.; Hadaschik, B.; Grunwald, V. (2023). Metastatic Prostate Cancer—A Review of Current Treatment Options and Promising New Approaches. Cancers (Basel), 15(2).
Removed references:
Schmid, S. et al. (2020). Activity of Platinum-Based Chemotherapy in Patients With Advanced Prostate Cancer With and Without DNA Repair Gene Aberrations. JAMA Netw Open, 3(10), e2021692.
Ruiz de Porras, V.; Font, A.; Aytes, A. (2021). Chemotherapy in Metastatic Castration-Resistant Prostate Cancer: Current Scenario and Future Perspectives. Cancer Lett, 523, 162–169.
Ruiz de Porras, V. et al. (2016). Curcumin Mediates Oxaliplatin-Acquired Resistance Reversion in Colorectal Cancer Cell Lines Through Modulation of CXC-Chemokine/NF-kappaB Signalling Pathway. Sci Rep, 6, 24675.
We have retained references [33, 34, 35, 36, and 43], as these correspond to the publicly available RNA-seq datasets analyzed in this study and are therefore essential to the validity and reproducibility of our results. These datasets encompass benign prostate, localized PC, CRPC, and NEPC tissues and form the foundation for the transcriptomic analyses of survivin (BIRC5) expression presented in the manuscript.
Reviewer Comment 7:
Overall, in the present state, the results cannot support the conclusions.
Authors Response:
We appreciate the reviewer’s thoughtful assessment and understand the concern regarding the strength of the conclusions. Our study was designed as a translational and hypothesis-generating investigation focused on linking clinical outcomes with biological mechanisms. Accordingly, our conclusions emphasize the integration of clinical, molecular, and functional data rather than definitive therapeutic claims.
In summary, we found that (1) high BIRC5 expression was associated with poorer overall and progression-free survival and reduced PSA response in platinum-treated mCRPC patients; (2) BIRC5 was enriched in AR-independent and NEPC-like contexts; and (3) survivin inhibition via YM155 synergized with carboplatin in AR-negative PC cell models. These concordant findings provide biological and translational plausibility supporting survivin as a potential biomarker and therapeutic target in platinum-treated, AR-independent mCRPC.
In line with the reviewer’s comments, we have carefully revised several sections of the manuscript to clarify the novelty, scope, and limitations of our findings. The Introduction and Discussion were expanded to better differentiate our work from previous studies and to emphasize that our results are exploratory and context-specific. The Abstract and Conclusions were also refined to ensure that the claims are appropriately measured and fully supported by the presented evidence:
Abstract; Page 2; Line 50: Our findings suggest that survivin may serve as a prognostic biomarker and potential therapeutic target in platinum-treated, AR-independent mCRPC. The integration of clinical and functional data provides translational support for combining the survivin inhibitor YM155 with platinum-based therapy. These results warrant further valida-tion in larger patient cohorts and in vivo models.
Page 13, Line 347: In summary, our findings suggest that survivin may serve as a prognostic biomarker and potential therapeutic target in platinum-treated, AR-independent mCRPC. The preclinical data demonstrate that YM155 effectively suppresses survivin expression and enhances the cytotoxic activity of carboplatin, providing translational support for this combination strategy. By integrating molecular profiling, clinical outcomes, and functional assays, this work highlights survivin as a biologically relevant target in mCRPC. Given the exploratory and hypothesis-generating scope of this work, further validation in larger patient cohorts and in vivo models will be essential to confirm these findings and define their clinical relevance.
Reviewer 2 Report
Comments and Suggestions for Authors
The article discusses the synergistic effect of Sepantronium bromide (YM155) in combination with carboplatin and some other known antitumor agents against certain resistant prostate cancer, and it recorded an enhancement of the tumor's response in the case of the proposed combination.
However, it doesn't deal with a novel structure. The article is interesting and provides a new perspective on cancer treatment.
However, a few remarks should be addressed to enhance the quality of the article; hence, I recommend the following:
1- The authors briefly described the chemical structure of YM155, although it is the core of their article. They should add the chemical structure of this compound.
2- Figures 1, 3, and 4 have very long legends. I advise separating the merged figure into more than one and annotating each with its suitable legend for readability and comprehensive improvement.
3- I would add the suggested mechanism of action and mode of binding of YM155 toward survivin if available in the literature,
see: Int J Biochem Mol Biol. 2012;3(2):179-97 and cancer cells. Sci Rep 12, 13135 (2022).
Author Response
Dear Reviewer,
We sincerely thank you for your careful reading of our manuscript and your constructive comments. We have revised the manuscript thoroughly in response to your suggestions, which have helped us to significantly improve the clarity, accuracy, and completeness of our review. All modifications in the manuscript have been made using the track changes function. Below we provide a detailed, point-by-point response to each of your comments.
Reviewer Comment 1:
The authors briefly described the chemical structure of YM155, although it is the core of their article. They should add the chemical structure of this compound
Authors Response:
We thank the reviewer for this helpful suggestion. In line with the reviewer’s comment, we have now included the chemical structure of YM155 (Sepantronium bromide) as Supplementary Figure 1 to provide a clearer visualization of the compound discussed throughout the manuscript.
Reviewer Comment 2:
Figures 1, 3, and 4 have very long legends. I advise separating the merged figure into more than one and annotating each with its suitable legend for readability and comprehensive improvement.
Authors Response:
We thank the reviewer for this valuable suggestion aimed at improving the clarity and readability of the figures. We fully agree that concise figure legends facilitate interpretation; however, we believe that the current organization of the results into four main figures provides an optimal balance between clarity and conceptual coherence.
Specifically, the figures have been structured to reflect the four key experimental and analytical blocks on which the manuscript is based:
- Figure 1: Analysis of survivin (BIRC5) expression across publicly available datasets representing different stages of prostate cancer, demonstrating that survivin is overexpressed in mCRPC and NEPC and correlates with shorter overall survival.
- Figure 2: Correlation between survivin expression and clinical outcomes in platinum-treated mCRPC patients.
- Figure 3: Functional evaluation of the effects of YM155 monotherapy on survivin expression, cell viability, clonogenic capacity, and apoptosis in DU145 and PC3 cells.
- Figure 4: Functional analysis of the combined effects of YM155 and carboplatin in AR-negative mCRPC cell models (DU145 and PC3).
Each figure, therefore, corresponds to a distinct and logically connected section of the study, ensuring that the flow of results is clear and that each experimental question is supported by a single integrated figure. We have, nonetheless, reviewed and refined the legends to improve readability and to ensure consistency in formatting and terminology.
Reviewer Comment 3:
I would add the suggested mechanism of action and mode of binding of YM155 toward survivin if available in the literature, see: Int J Biochem Mol Biol. 2012;3(2):179-97 and cancer cells. Sci Rep 12, 13135 (2022).
Authors Response:
We thank the reviewer for this insightful suggestion. We agree that including a brief summary of the proposed mechanism of action of YM155 toward survivin adds valuable mechanistic context. Accordingly, we have incorporated the articles suggested by the reviewer (Int J Biochem Mol Biol., 2012; Sci Rep., 2022) and included a concise description of YM155’s proposed mechanisms of action in the Introduction to provide additional background and to better contextualize the rationale for survivin inhibition in our study:
Page 3; Line 91: YM155 is a cell-permeable imidazolium derivative that suppresses BIRC5 transcription by directly binding to its promoter region and disrupting transcriptional activation survivin expression by directly binding to its promoter [23,24]. Recent studies have revealed that YM155 induces DNA damage and activates cellular stress pathways, including modulation of PI3K/AKT and mTOR signaling. Moreover, YM155 localizes to mitochondria, where it promotes mitochondrial dysfunction, increases glycolytic intermediates, and decreases oxidative phosphorylation and tricarboxylic acid (TCA) cycle activity, ultimately triggering AMP-activated kinase (AMPK) activation [25].
Reviewer 3 Report
Comments and Suggestions for Authors
The manuscript entitled “YM155 Inhibition of Survivin Enhances Carboplatin Efficacy in Metastatic Castration-Resistant Prostate Cancer” presents relevant results and is generally well written. However, specific areas requiring adjustments in wording and methodological precision have been identified. Specific observations based on a critical review of the content are detailed below.
Abstract
The authors wrote the following: “Methods: RNAseq analysis of tissue samples from mCRPC patients was performed to assess correlations between survivin expression and clinical outcomes”.’ However, in the materials and methods section, I did not see any section that considers the RNSseq method.
It is recommended that the name of the compound YM155: Sepantronium bromide, be mentioned in the abstract and introduction.
Materials and Methods
The wording of section 4.1. Computational analysis of human PC data should be improved. It is advisable to indicate the meaning of the abbreviations FPKM and HTSeg. When referring to software, the version used should be indicated, as well as the brand name and country of origin.
Why did the authors of this study select the DU145 cell line? Are there other types of prostate cancer cells? What was the basis for their selection?
In relation to method 4.5. Cytotoxicity of drug combinations, briefly describe the MTT assay or indicate the bibliographic reference used for this procedure.
Line 350: The authors mention the use of a serial dilution of DU145 and PC3. My question is: Is every PC3 cell line castration-resistant? Or should the name PC3 mCRPC be used?
In section 4.6. Colony formation assay, the authors do not mention the different types of dilutions they used for YM155 and carboplatin. I recommend that they indicate this.
With regard to the materials, equipment, reagents, and software mentioned in the Materials and Methods section, the following is recommended: A) Reagents: Indicate the catalogue number, brand name, and country of origin in parentheses. B) Equipment: Specify the name and model of the equipment, followed in parentheses by the brand name and country of origin. C) Software: Mention the name and version used, including the brand name and country of origin in parentheses.
Line 261: Is the AR-negative mCRPC cell line the same as the PC3 cell line and the same as the mCRPC cell line?
Results
In the statistical analysis section, the authors mention that they used various statistical tests, therefore, it is important that the type or types of statistical tests used are indicated in the legend of each figure. For example, in the legend of Figure 1, the authors do not mention which statistical test they used to calculate the p-value in Figure 1B.
In the legend for Figure 4, the authors do not indicate the name of the statistical tests used to quantify the p-value.
The meaning of the abbreviations used should be indicated in the legend for each figure.
The image in Figure 3 does not show the name of the variable YM155 in Figures A and C.
In the legend of Figure 4, the authors do not indicate the name of the statistical tests used to quantify the p-value.
Discussion
It is very important that the authors clearly mention the original contributions of this study and the potential therapeutic relevance of its results. A search for scientific articles in PubMed, using the words (survivin[Title]) AND (prostate cancer[Title]) AND (YM155[Title]) in the title of an article, yielded only five articles, four of which were not considered by the authors of this study, but which may be relevant to the discussion.
Tolcher AW, Quinn DI, Ferrari A, Ahmann F, Giaccone G, Drake T, Keating A, de Bono JS. A phase II study of YM155, a novel small-molecule suppressor of survivin, in castration-resistant taxane-pretreated prostate cancer. Ann Oncol. 2012;23(4):968-73. doi: 10.1093/annonc/mdr353.
Wang Q, Chen Z, Diao X, Huang S. Induction of autophagy-dependent apoptosis by the survivin suppressant YM155 in prostate cancer cells. Cancer Lett. 2011;302(1):29-36. doi: 10.1016/j.canlet.2010.12.007.
Aoyama Y, Kaibara A, Takada A, Nishimura T, Katashima M, Sawamoto T. Population pharmacokinetic modeling of sepantronium bromide (YM155), a small molecule survivin suppressant, in patients with non-small cell lung cancer, hormone refractory prostate cancer, or unresectable stage III or IV melanoma. Invest New Drugs. 2013;31(2):443-51. doi: 10.1007/s10637-012-9867-x.
Danielpour D, Gao Z, Zmina PM, Shankar E, Shultes BC, Jobava R, Welford SM, Hatzoglou M. Early Cellular Responses of Prostate Carcinoma Cells to Sepantronium Bromide (YM155) Involve Suppression of mTORC1 by AMPK. Sci Rep. 2019;9(1):11541. doi: 10.1038/s41598-019-47573-y. Erratum in: Sci Rep. 2019;9(1):14826. doi: 10.1038/s41598-019-51007-0.
Author Response
Dear Reviewer,
We sincerely thank you for your careful reading of our manuscript and your constructive comments. We have revised the manuscript thoroughly in response to your suggestions, which have helped us to significantly improve the clarity, accuracy, and completeness of our review. All modifications in the manuscript have been made using the track changes function. Below we provide a detailed, point-by-point response to each of your comments.
Abstract
Reviewer Comment 1:
“The authors wrote the following: “Methods: RNAseq analysis of tissue samples from mCRPC patients was performed to assess correlations between survivin expression and clinical outcomes”.’ However, in the materials and methods section, I did not see any section that considers the RNSseq method”
Authors Response:
We thank the reviewer for this valuable observation. We would like to clarify that no new RNA-Seq experiments were performed specifically for this study. Instead, we analyzed previously published RNA-Seq datasets generated by the group of Prof. Himisha Beltran (co-author of this manuscript) and collaborators.
For the analysis of survivin (BIRC5) expression in clinical prostate cancer samples, we used publicly available datasets encompassing benign prostate, localized PC, CRPC, and NEPC tissues, as reported in our references [30, 31, and 33]. These datasets were previously published by Prof. Beltran’s group and are explicitly cited in the Results section:
Page 3; Line 123: “RNA-Seq analysis of survivin (BIRC5) mRNA expression levels in datasets of benign prostate, localized PC, CRPC, and NEPC tissue samples [33, 34].”
Page 6, Line 166: “Given that platinum chemotherapy is sometimes used in mCRPC with AVPC features including NEPC, we evaluated the clinical associations of survivin (BIRC5) expression levels in platinum-treated patients (n = 23) [36].”
For the analysis of BIRC5 expression in preclinical prostate cancer models, we used RNA-Seq datasets from a previously published study [35], as described in the Results section:
Page 3, Line 132: “Consistent with these results, survivin (BIRC5) expression varied significantly across prostate cancer subtypes (ANOVA p = 0.049) among previously profiled preclinical models [35].”
We have now revised the Abstract and the Materials and Methods sections to explicitly specify that these RNA-Seq analyses were conducted using previously published datasets rather than through new sequencing experiments:
Abstract; page1; Line 34: Analysis of publicly available RNA-seq datasets from mCRPC patients was performed to assess correlations between survivin expression and clinical outcomes
Page 13; Line 349: We analyzed previously published RNA-Seq datasets generated by the group of Dr. Himisha Beltran and collaborators to assess mRNA expression levels for BIRC5, neuroendocrine genes, and AR/luminal markers across prostate cancer subtypes (34 benign, 68 localized PC, 31 CRPC, and 22 NEPC) [33, 34, 36]. Sequencing reads were aligned to the human reference genome (hg19/GRCh37) using STAR software (v2.3.0e; Cold Spring Harbor Laboratory, NY, USA). For each sample, HTSeq (v0.6.1; Python framework developed at EMBL, Heidelberg, Germany) was employed to generate read counts, while Cufflinks (v2.2.1; Trapnell Lab, University of Washington, Seattle, WA, USA) was used to calculate FPKM (Fragments Per Kilobase of transcript per Million mapped reads) values. In addition, we analyzed RNA-Seq data from previously pub-lished preclinical models [35] and clinical samples [43].
Reviewer Comment 2:
It is recommended that the name of the compound YM155: Sepantronium bromide, be mentioned in the abstract and introduction.
Authors Response:
We appreciate the reviewer’s suggestion. We agree that including the full chemical name of YM155 improves clarity and precision for readers who may be less familiar with the compound. Accordingly, we have revised both the Abstract and the Introduction to include the chemical name “Sepantronium bromide” when first mentioning YM155.
Moreover, in line with the reviewer’s comment and following a related suggestion from another reviewer, we have also included the chemical structure of YM155 as Supplementary Figure 1 to further enhance the characterization of this compound.
The following changes have been made in the revised manuscript:
Abstract; Page 1; Line 30: YM155 (Sepantronium bromide) suppresses survivin expression and has demonstrated antitumor activity in preclinical models.
Introduction; Page 3; Line 89: Several experimental strategies have been developed to target surviving [8], including YM155 (Sepantronium bromide), a small-molecule inhibitor of survivin ex-pression (Supplementary Figure 1).
Material and Methods
Reviewer Comment 3:
“The wording of section 4.1. Computational analysis of human PC data should be improved. It is advisable to indicate the meaning of the abbreviations FPKM and HTSeg. When referring to software, the version used should be indicated, as well as the brand name and country of origin”
Authors Response:
We thank the reviewer for this valuable and detailed suggestion. We have revised Section 4.1 (Computational analysis of human PC data) to improve clarity and completeness, and we have incorporated methodological details consistent with the procedures reported in the referenced studies.
Specifically, we have:
- Clarified the meaning of the abbreviations FPKM (Fragments Per Kilobase of transcript per Million mapped reads) and HTSeq (High-Throughput Sequencing framework).
- Added detailed information on the software versions, developers, and country of origin (STAR, HTSeq, and Cufflinks).
- Refined the overall wording of the section to enhance readability and methodological precision.
The revised section now reads as follows:
Page 14; Line 372: We analyzed previously published RNA-Seq datasets generated by the group of Dr. Himisha Beltran and collaborators to assess mRNA expression levels for BIRC5, neuroendocrine genes, and AR/luminal markers across prostate cancer subtypes (34 benign, 68 localized PC, 31 CRPC, and 22 NEPC) [33, 34, 36]. Sequencing reads were aligned to the human reference genome (hg19/GRCh37) using STAR software (v2.3.0e; Cold Spring Harbor Laboratory, NY, USA). For each sample, High-Throughput Se-quencing framework (HTSeq) v0.6.1 (Python framework developed at EMBL, Heidel-berg, Germany) was employed to generate read counts, while Cufflinks (v2.2.1; Trap-nell Lab, University of Washington, Seattle, WA, USA) was used to calculate FPKM (Fragments Per Kilobase of transcript per Million mapped reads) values. In addition, we analyzed RNA-Seq data from previously published preclinical models [35] and clinical samples [43].
Reviewer Comment 4:
Why did the authors of this study select the DU145 cell line? Are there other types of prostate cancer cells? What was the basis for their selection?
Authors response:
We thank the reviewer for this relevant and insightful question. Our study focuses on the metastatic castration-resistant prostate cancer setting. Therefore, the selected models—PC3 and DU145—are among the most widely used and well-characterized androgen-insensitive prostate cancer cell lines that reflect this disease stage.
The DU145 cell line was derived from a brain metastasis, while PC3 originated from a bone metastasis of prostate cancer patients. Both cell lines lack androgen receptor (AR) expression and are non-responsive to androgen stimulation, thereby representing models of androgen-independent, castration-resistant, and metastatic prostate cancer (Saranyutanon et al., Cancers 2020, 12(9), 2653; Sobel & Sadar, J Urol 2005, 173(2), 342–359).
These models are extensively used in preclinical research to study molecular mechanisms of castration resistance and to evaluate novel therapeutic strategies in the advanced disease context. Accordingly, their use in our study allowed us to evaluate the effects of survivin inhibition and platinum-based therapy under conditions that closely mimic androgen-insensitive metastatic prostate cancer.
To clarify this rationale, we have added the following sentence to the Materials and Methods (Section 4.2, Cell lines and culture conditions):
Page 14, Line 362: DU145 and PC3 cell lines were selected as representative models of androgen-insensitive metastatic prostate cancer, widely recognized as models of mCRPC [44].
Reviewer Comment 5:
In relation to method 4.5. Cytotoxicity of drug combinations, briefly describe the MTT assay or indicate the bibliographic reference used for this procedure.
Authors response:
We thank the reviewer for this observation. We believe this may stem from a misunderstanding. The MTT assay is already described in detail in Section 4.4. Cell Viability Assay, where we provide the methodology and reference for the procedure used to assess cell viability.
Section 4.5. Cytotoxicity of drug combinations, on the other hand, specifically describes the experimental design used to evaluate drug synergism between YM155 and carboplatin according to the Chou–Talalay method.
To prevent any possible confusion, we have now added a brief clarifying sentence at the beginning of Section 4.5 indicating that the MTT assay described in Section 4.4 was used to determine cell viability for the combination studies.
Page 15, Line 413: Cell viability for drug combination studies was determined using the MTT assay described in Section 4.4, and drug interactions were analyzed according to the Chou–Talalay method.
Reviewer Comment 6:
Line 350: The authors mention the use of a serial dilution of DU145 and PC3. My question is: Is every PC3 cell line castration-resistant? Or should the name PC3 mCRPC be used?
Authors response:
We thank the reviewer for this thoughtful observation. In this section, the phrase “serial dilution of DU145 and PC3” refers to the serial dilution of the initial cell suspension used to seed 500 cells per well in 6-well plates for the colony formation assay, rather than to any modification of the cell lines themselves.
As described in our response to the previous comment, the PC3 cell line was derived from a bone metastasis of a prostate cancer patient and lacks androgen receptor expression. It is non-responsive to androgen stimulation, thereby representing a model of androgen-independent, castration-resistant, and metastatic prostate cancer (Sobel & Sadar, J Urol 2005; Saranyutanon et al., Cancers 2020).
Based on these biological characteristics, PC3 is widely recognized and used as a representative model of mCRPC. Accordingly, we have clarified this concept in the Materials and Methods (Section 4.2, Cell lines and culture conditions) to ensure terminological precision:
Page 14, Line 362: DU145 and PC3 cell lines were selected as representative models of androgen-insensitive metastatic prostate cancer, widely recognized as models of mCRPC [44].
Reviewer Comment 7:
In section 4.6. Colony formation assay, the authors do not mention the different types of dilutions they used for YM155 and carboplatin. I recommend that they indicate this.
Authors Response:
We thank the reviewer for this helpful comment. The concentrations of YM155 and carboplatin used in the colony formation assays are already indicated in Figure 3, and the concentrations used in the drug combination experiments are shown in Figure 4. However, to improve clarity and ensure that the methodological description is self-contained, we have now included these details explicitly in Section 4.6 (Colony formation assay):
Page 15; Line 429: For monotherapy experiments, YM155 was tested at concentrations ranging from 2.5 to 40 nM for DU145 (2.5, 5, 10, 20, and 40 nM) and from 0.5 to 10 nM for PC3 (0.5, 1, 2.5, 5, and 10 nM). For drug combination experiments, the following concentrations were used: DU145 – carboplatin 0.5 µM and YM155 10 nM; PC3 – carboplatin 2.5 µM and YM155 2.5 nM.
Reviewer Comment 8:
With regard to the materials, equipment, reagents, and software mentioned in the Materials and Methods section, the following is recommended: A) Reagents: Indicate the catalogue number, brand name, and country of origin in parentheses. B) Equipment: Specify the name and model of the equipment, followed in parentheses by the brand name and country of origin. C) Software: Mention the name and version used, including the brand name and country of origin in parentheses.
Authors Response:
We thank the reviewer for this valuable and detailed suggestion. In accordance with the journal’s reporting standards, we have carefully revised the Materials and Methods section to include detailed information about the reagents, equipment, and software used in this study.
Specifically, we have:
- Added catalogue numbers, brand names, and country of origin for all key reagents.
- Specified the name, model, brand, and country of origin for all equipment.
- Included the name, version, and developer/country for all software used.
These details have been incorporated throughout the Materials and Methods section to enhance transparency and reproducibility.
Reviewer Comment 9:
Line 261: Is the AR-negative mCRPC cell line the same as the PC3 cell line and the same as the mCRPC cell line?
Authors Response:
We thank the reviewer for this observation. We understand that the previous wording might have caused confusion regarding the identity of the AR-negative mCRPC cell line.
To clarify, in our study the AR-negative mCRPC cell lines refer specifically to the DU145 and PC3 models. These are two of the most widely used and well-characterized androgen-insensitive prostate cancer cell lines that represent the metastatic castration-resistant prostate cancer stage.
The DU145 cell line was derived from a brain metastasis, whereas PC3 originated from a bone metastasis of prostate cancer patients. Both cell lines lack androgen receptor expression and are non-responsive to androgen stimulation, thereby serving as representative models of androgen-independent, castration-resistant, and metastatic prostate cancer (Saranyutanon et al., Cancers 2020, 12(9), 2653; Sobel & Sadar, J. Urol. 2005, 173(2), 342–359).
To avoid ambiguity, we have now revised the text to explicitly specify the cell lines:
Page 11; Line 276: Importantly, the combination of YM155 with carboplatin exhibited a synergistic effect in DU145 and PC3 AR-negative mCRPC cell lines
Results
Reviewer Comment 10:
In the statistical analysis section, the authors mention that they used various statistical tests, therefore, it is important that the type or types of statistical tests used are indicated in the legend of each figure. For example, in the legend of Figure 1, the authors do not mention which statistical test they used to calculate the p-value in Figure 1B. In the legend for Figure 4, the authors do not indicate the name of the statistical tests used to quantify the p-value.
Authors Response:
We thank the reviewer for this valuable and precise observation. We fully agree that the statistical tests used should be clearly indicated in the figure legends to ensure transparency, reproducibility, and consistency with the Statistical Analyses section.
Accordingly, we have carefully reviewed and updated all figure legends to specify the statistical test(s) applied in each case:
- Figure 1B: p-values were calculated using one-way ANOVA followed by Welch’s t-test.
- Figure 2: Survival analyses were performed using the Kaplan–Meier method and compared using the log-rank test.
- Figures 3 and 4: Statistical significance between treatment groups was determined using a two-tailed Student’s t-test.
We also ensured full alignment between the Statistical Analyses section (Section 4.9) and all figure legends.
Reviewer Comment 11:
The meaning of the abbreviations used should be indicated in the legend for each figure.
Authors Response:
We thank the reviewer for this helpful and precise suggestion. We fully agree that all abbreviations used in the figures should be defined within each legend to ensure clarity and facilitate independent interpretation of the results.
Accordingly, we have reviewed and updated the legends of all figures (1–4) to include the full meaning of each abbreviation at the end of the legend.
Reviewer Comment 12:
The image in Figure 3 does not show the name of the variable YM155 in Figures A and C
Authors Response:
We thank the reviewer for this observation and appreciate the opportunity to clarify this point. All experiments shown in Figure 3 were performed using YM155 as a single agent, as indicated in the figure title and throughout the legend. For this reason, we did not include the drug name within individual subpanels (A–D), to avoid redundancy and maintain visual clarity.
Nevertheless, we have carefully reviewed the figure and confirmed that the experimental context (YM155 monotherapy) is clearly described in both the title (“Effects of YM155 on survivin expression, cell viability, colony formation, and cell death...”) and the corresponding legend. Therefore, we believe additional labelling within the panels is unnecessary and could compromise figure readability.
Reviewer Comment 13:
In the legend of Figure 4, the authors do not indicate the name of the statistical tests used to quantify the p-value.
Authors Response:
We thank the reviewer for this helpful remark. We have now specified the statistical test used in the legend of Figure 4, to ensure clarity and consistency with the Statistical Analyses section.
The updated legend includes the following sentence:
“Statistical significance between treatment groups was determined using a two-tailed Student’s t-test.”
Discussion
Reviewer Comment 14:
“It is very important that the authors clearly mention the original contributions of this study and the potential therapeutic relevance of its results. A search for scientific articles in PubMed, using the words (survivin[Title]) AND (prostate cancer[Title]) AND (YM155[Title]) in the title of an article, yielded only five articles, four of which were not considered by the authors of this study, but which may be relevant to the discussion.
Tolcher AW, Quinn DI, Ferrari A, Ahmann F, Giaccone G, Drake T, Keating A, de Bono JS. A phase II study of YM155, a novel small-molecule suppressor of survivin, in castration-resistant taxane-pretreated prostate cancer. Ann Oncol. 2012;23(4):968-73. doi: 10.1093/annonc/mdr353.
Wang Q, Chen Z, Diao X, Huang S. Induction of autophagy-dependent apoptosis by the survivin suppressant YM155 in prostate cancer cells. Cancer Lett. 2011;302(1):29-36. doi: 10.1016/j.canlet.2010.12.007.
Aoyama Y, Kaibara A, Takada A, Nishimura T, Katashima M, Sawamoto T. Population pharmacokinetic modeling of sepantronium bromide (YM155), a small molecule survivin suppressant, in patients with non-small cell lung cancer, hormone refractory prostate cancer, or unresectable stage III or IV melanoma. Invest New Drugs. 2013;31(2):443-51. doi: 10.1007/s10637-012-9867-x.
Danielpour D, Gao Z, Zmina PM, Shankar E, Shultes BC, Jobava R, Welford SM, Hatzoglou M. Early Cellular Responses of Prostate Carcinoma Cells to Sepantronium Bromide (YM155) Involve Suppression of mTORC1 by AMPK. Sci Rep. 2019;9(1):11541. doi: 10.1038/s41598-019-47573-y. Erratum in: Sci Rep. 2019;9(1):14826. doi: 10.1038/s41598-019-51007-0”
Authors Response:
We sincerely thank the reviewer for this thoughtful and constructive comment. We fully agree that it is essential to clarify both the original contributions of our study and its therapeutic relevance, as well as to acknowledge and contextualize previous publications on survivin and YM155 in prostate cancer.
We acknowledge that survivin/BIRC5 and YM155 are not conceptually new. The novelty of our work lies in its translational and context-specific evaluation of survivin in androgen receptor (AR)-independent and neuroendocrine-like metastatic castration-resistant prostate cancer (mCRPC), particularly in patients treated with platinum chemotherapy — a clinical setting that, to our knowledge, has not been previously linked to BIRC5 expression or YM155 efficacy.
Specifically, our study provides:
- Clinical evidence showing that high BIRC5 expression correlates with poorer overall and progression-free survival, as well as reduced PSA response, in platinum-treated mCRPC patients.
- Biological positioning of survivin within AR-low/NEPC-like prostate cancer states.
- Functional validation that inhibition of survivin with YM155 synergistically enhances carboplatin efficacy in AR-negative mCRPC cell models.
In response to the reviewer’s suggestion, we have incorporated the articles proposed by the reviewer—Wang et al., 2011; Tolcher et al., 2012; Aoyama et al., 2013; and Danielpour et al., 2019—into the Discussion section to ensure a more complete contextualization of YM155’s previously reported mechanisms and clinical studies. These references complement our findings and reinforce the novelty of evaluating survivin inhibition and YM155–platinum synergy in AR-independent mCRPC.
These studies primarily addressed taxane resistance or non-platinum contexts and did not evaluate survivin as a predictive biomarker or test the combination of YM155 with platinum compounds in AR-negative models. Our work extends this knowledge by integrating clinical, molecular, and functional data to demonstrate that survivin inhibition can enhance platinum efficacy in AR-independent mCRPC.
We have also cited Yu et al., 2025, who identified BIRC5 and BIRC7 as potential prognostic biomarkers through bioinformatic analyses. While their study did not include platinum-treated patients or experimental validation, our findings complement and extend theirs by showing that BIRC5 expression correlates with platinum response and that YM155 sensitizes AR-negative mCRPC cells to carboplatin.
Our manuscript therefore does not claim the first discovery of survivin or YM155 but provides new clinical–translational evidence supporting survivin/BIRC5 as a context-specific biomarker and therapeutic target in platinum-treated, AR-independent mCRPC.
To further emphasize these aspects, we have revised the Introduction and Discussion as follows:
Page 3; Line 92: Although previous studies have explored the inhibition of survivin in PC, most have focused on restoring sensitivity to taxane-based therapies, such as docetaxel or cabazitaxel [29, 30]. Recent bioinformatic analyses, including the work by Yu et al. (2025), have also identified members of the BIRC family—particularly BIRC5 and BIRC7—as potential prognostic and therapeutic biomarkers in PC [32]. However, the potential role of survivin expression in shaping response to platinum-based chemotherapy has not been systematically evaluated in mCRPC. Considering that AR-independent and neuroendocrine-like prostate cancers are more likely to receive platinum treatment, our study specifically addresses the clinical and biological relevance of BIRC5/survivin in this disease context.
Page 12; Line 288: In line with these observations, Wang et al. (2011) reported that YM155 triggers autophagy-dependent apoptosis in prostate cancer cells [39], while Danielpour et al. (2019) demonstrated that it suppresses mTORC1 activity via AMPK activation, reinforcing its role as a broader stress-response modulator beyond survivin inhibition [40].
Clinically, Tolcher et al. (2012) conducted a phase II clinical trial in taxane-pretreated CRPC patients showing modest single-agent activity but acceptable tolerability [41], and Aoyama et al. (2013) described its pharmacokinetic profile, supporting further exploration in combination strategies [42].
Building on this evidence, our study provides new functional and clinical data in a distinct context—platinum-treated, AR-negative mCRPC—showing that YM155 synergistically enhances carboplatin efficacy. This synergy suggests that survivin inhibition may sensitize resistant prostate cancer cells to DNA-damaging agents, offering a rationale for the biomarker-driven evaluation of this combination.
Page 12; Line 302: Whereas previous reports mainly investigated YM155 in combination with tax-anes to overcome chemoresistance in PC models [30] or identified survivin family members such as BIRC5 and BIRC7 as prognostic biomarkers through bioinformatic analyses [32], our work extends these findings by linking survivin ex-pression to platinum response and by functionally validating the YM155–carboplatin synergy in AR-negative mCRPC models.